# RDR: the Recap, Deliberate, and Respond Method for Enhanced Language Understanding

**Yuxin Zi**[1*], **Hariram Veeramani**[2*], **Kaushik Roy**[1], **Amit Sheth**[1]

[1]Artificial Intelligence Institute, University of South Carolina, Columbia, SC, USA
[2]University of California, Los Angeles (UCLA)
yzi@email.sc.edu, hariram@ucla.edu, kaushikr@email.sc.edu, amit@sc.edu

## Abstract

Natural language understanding (NLU) using neural network pipelines often requires additional context that is not solely present in the input data, such as external knowledge graphs. Through prior research, it has been evident that NLU benchmarks are susceptible to manipulation by neural models - these models exploit statistical artifacts within the encoded external knowledge to artificially inflate performance metrics for downstream tasks. Our proposed approach, known as the Recap, Deliberate, and Respond (RDR) paradigm, addresses this issue by incorporating three distinct objectives within the neural network pipeline. The Recap objective involves paraphrasing the input text using a paraphrasing model in order to summarize and encapsulate salient information of the input. Deliberate refers to encoding the external graph information that is relevant to entities in the input text using a graph embedding model. Finally, Respond employs a classification head model that integrates representations from the Recap and Deliberate steps to generate the final prediction. By cascading these three models and minimizing a combined loss, we mitigate the potential of the model gaming the benchmark, while establishing a robust method for capturing the underlying semantic patterns to achieve accurate predictions. We conduct tests on multiple GLUE benchmark tasks to evaluate the effectiveness of the RDR method. Our results demonstrate improved performance compared with competitive baselines, with an enhancement of up to 2% on standard evaluation metrics. Furthermore, we analyze the observed behavior of semantic understanding of the RDR models, emphasizing their ability to avoid gaming the benchmark while accurately capturing the true underlying semantic patterns.

## Introduction

Previous research in the field of natural language understanding (NLU) and neural network pipelines has acknowledged the necessity of incorporating additional context beyond the input data (Sheth et al. 2021). To address this limitation, one well-established approach is to integrate external knowledge graphs as supplementary context (Zhu et al. 2023). These knowledge graphs contain structured information about entities, relationships, and concepts. This external information enables the neural network to infer semantic

*These authors contributed equally to this work.

connections between entities, even when such relations are not explicitly stated in the data alone. Thus, the neural network is able to uncover implicit or missing contextual associations.

However, a notable concern has been raised by previous research on the vulnerability of NLU benchmarks when facing manipulation by neural models (Bender and Koller 2020; McCoy, Pavlick, and Linzen 2019). This issue casts doubt on the reliability and generalizability of the performance metrics reported by neural models on benchmark datasets, and has undergone extensive scrutiny within the NLU community. Researchers have explored various methods to mitigate benchmark gaming to ensure the NLU models exhibit authentic language understanding. These methods include introducing more comprehensive evaluation protocols, employing adversarial testing, and integrating external knowledge into training. However, the first two approaches do not redesign the training pipeline to enhance the model's language understanding capability. Although the knowledge integration method does modify the training procedure, it remains unclear whether the external knowledge is factually and sufficiently enforced into the integration process.

We propose a novel approach called the Recap, Deliberate, and Respond (RDR) paradigm, which addresses these limitations by integrating three distinct objectives within the neural network pipeline. The first objective, Recap, involves paraphrasing the input text using a dedicated model. This process captures the essential and salient information in the input. The second objective, Deliberate, focuses on encoding external graph information that relates to the entities appeared in the input text. This step utilizes a graph embedding model to leverage the knowledge within the knowledge graphs. By integrating this external context into the neural network pipeline, the Deliberate objective enhances the model's capability of comprehending relationships between entities and extract relevant information for downstream tasks. The final objective in the RDR paradigm is Respond, which employs a classification head model. This model utilizes representations from the Recap and Deliberate modules to generate the final prediction. By incorporating insights from the Recap and Deliberate stages, the Respond objective enables more accurate and informed predictions. The cascading structure of these three objectives, along with minimizing a combined loss, prevents the model

from artificially inflate performance metrics through exploiting statistical artifacts. Our robust methodology facilitates the capturing of the true underlying semantic patterns of the input data with the assistance of external knowledge, which lead to more reliable and accurate predictions.

To evaluate the effectiveness of the RDR paradigm, we test our method on multiple GLUE benchmark tasks that involve sentence similarity, textual entailment, and natural language inference (Wang et al. 2018; Sharma et al. 2019; Demszky, Guu, and Liang 2018; Dolan and Brockett 2005; Poliak 2020). The results demonstrate superior performance compared to competitive baselines, with improvements of up to 2% on standard evaluation metrics. These findings highlight the capability of the RDR approach to enhance NLU performance of neural models. Furthermore, with inference examples from the RDR models, we discuss the model's robustness for semantic understanding against statistical artifacts. We find that the introduction of the Recap and Deliberate objectives leads to better comprehension of the underlying semantic patterns in both data and external knowledge.

## RDR-Methodology

Figure 1 shows an overview of the traditional training pipeline for integrating external knowledge within neural networks, and our RDR method.

### Traditional Method
#### Notations: Functions, and their Inputs and Outputs

- *Input Text: $x$*, Tokenized Text: $T(x)$
- *Language Model, Input, Function and Output:* $x' = f(T(x), \theta)$
- *Subgraph Extractor, Input, Function, and Output:* $KG_x = subgraph\_extract(T(x), KG)$, here $KG$ is a large knowledge graph (e.g., ConceptNet).
- *Graph Embedding Model, Input, Function and Output:* $e_x = Aggr(g(KG_x, i \in KG_x, \theta'))$, here $Aggr$ is an aggregation function (e.g., average of all node embeddings in $KG_x$), $i \in KG_x$ denotes the nodes in subgraph $KG_x$.
- *Embedding Fusion Model with Classification Head, Input, Function, and Output:* $z = h(e_x, x', \theta'')$
- *Loss:* Cross Entropy (CE) loss with ground truth denoted as $y$, $CE(z, y)$.

#### Forward Pass and Loss Calculation During Training
The steps for the traditional method are as follows:

1. Feed the tokenized text $T(x)$ into a language model, obtaining the embedding $x'$.
2. Apply an *off-the-shelf* graph extraction method to extract a subgraph $KG_x$ from the larger knowledge graph $KG$.
3. Apply a graph embedding model $Aggr(g(KG_x, \theta'))$ to obtain the graph embedding for $x$, denoted as $e_x$.
4. Pass the language model embedding $x'$ and the subgraph embedding $e_x$ into an embedding fusion model with a classification head $h(e_x, x', \theta'')$ to obtain the logits $z$. Compute the loss using logits $z$ and ground truth $y$.

### The RDR Method
#### Notations: Functions, and their Inputs and Outputs

- *Input Text: $x$*, Tokenized Text: $T(x)$
- *Paraphrasing Model, Input, Function and Output:* $x' = f(T(x), \theta)$
- *Paraphrasing Loss:* The discrepancy between the paraphrased text $x'$ and the original text $x$ is measured using cross entropy loss between the logits from the model $f$ and the ground truth distribution of tokens in $x$. We denote this loss as $PL(x', x)$.
- *Subgraph Extractor, Input, Function, and Output:* $KG_x = subgraph\_extract(T(x), KG)$, here $KG$ is a large knowledge graph (e.g., ConceptNet).
- *Graph Embedding Model, Input, Function and Output:* $e_x = Aggr(g(KG_x, i \in KG_x, \theta'))$, here $Aggr$ is an aggregation function (e.g., average of all node embeddings in $KG_x$), $i \in KG_x$ denotes the nodes in subgraph $KG_x$.
- *Embedding Loss Calculator:* First, all links in $KG_x$ are predicted using the model $Aggr(g(KG_x, i \in KG_x, \theta'))$. We define a link between two nodes $i$ and $j$ to be exist if $||g(KG_x, i, \theta') - g(KG_x, j, \theta')|| \leq \tau$, where $tau$ is a "closeness" threshold set empirically. Then, we calculate link prediction metrics, e.g., mean reciprocal rank (MRR), hits@k, etc, and denote as $GEL(x, KG_x)$.
- *Embedding Fusion Model with Classification Head, Input, Function, and Output:* $z = h(e_x, x', \theta'')$
- *Response Loss:* Cross Entropy (CE) loss with ground truth denoted as $y$, $RL(z, y)$.
- *Total Loss:* $L = PL(x', x) + GEL(x, KG_x) + RL(z, y)$.

#### Forward Pass and Loss Calculation During Training
We describe the steps of our RDR method:

1. Fed the tokenized text $T(x)$ into a language model, obtaining the embedding $x'$. Calculate the paraphrasing loss $PL(x', x)$.
2. Apply an *off-the-shelf* graph extraction method to extract a subgraph $KG_x$ from the larger knowledge graph $KG$.
3. Apply a graph embedding model $Aggr(g(KG_x, \theta'))$ to obtain the graph embedding for $x$, denoted as $e_x$. Compute the link prediction loss as $GEL(x, KG_x)$.
4. Pass the language model embedding $x'$ and the subgraph embedding $e_x$ into an embedding fusion model with a classification head $h(e_x, x', \theta'')$ to obtain the logits $z$. Compute the response loss as $RL(z, y)$
5. Compute the total loss as $L = PL(x', x) + GEL(x, KG_x) + RL(z, y)$.

## Experiments and Results

Our implementation utilizes task-specific hyperparameters, previously identified as optimal for the GLUE benchmark, except that we train for 1 epoch instead of 3. Throughout the training and evaluation process, a batch size of 8 is employed. We also integrate task-specific knowledge graphs or subgraph representations, amounting to up to 10% of the total knowledge graph triples. Our approach adheres

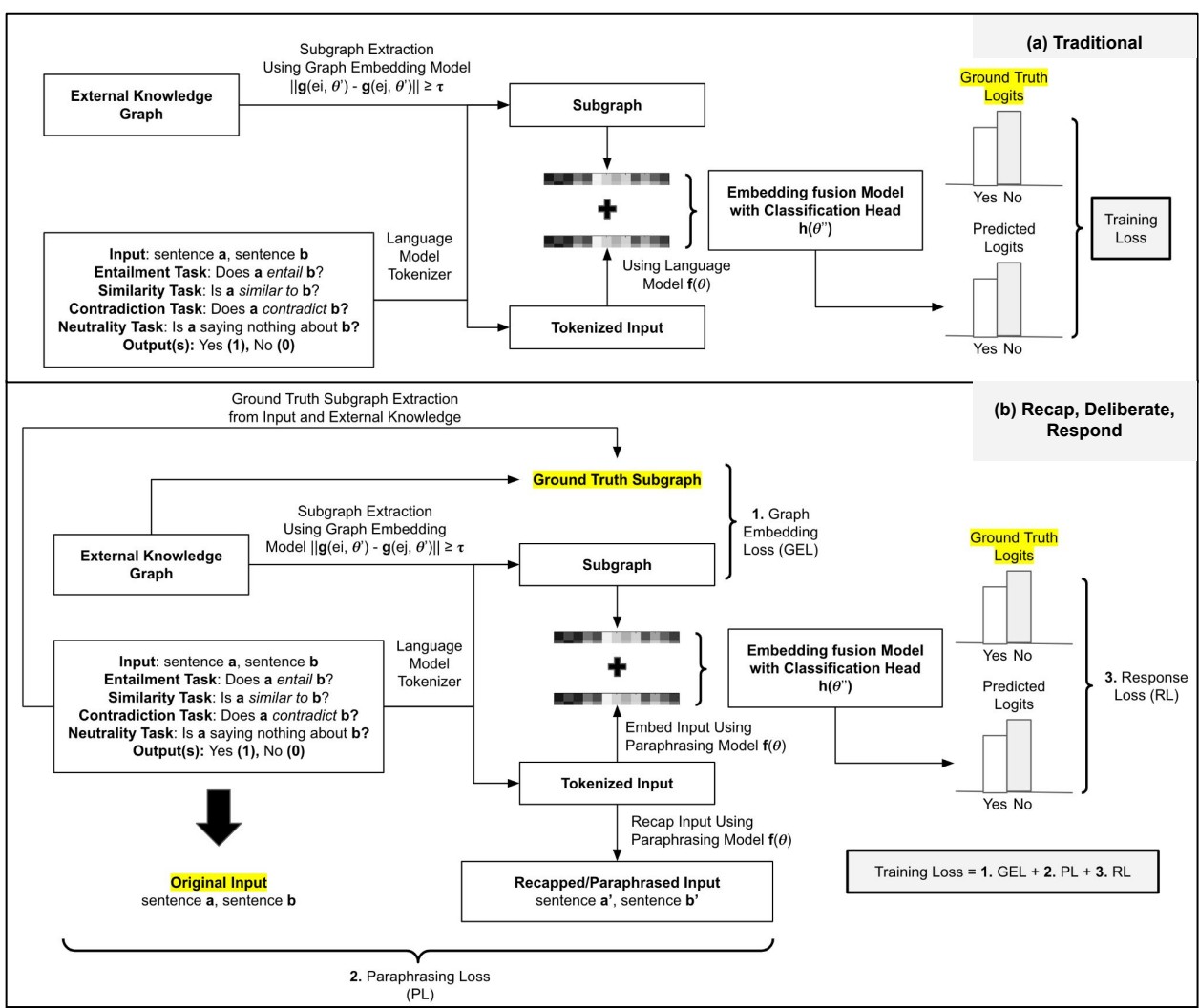

Figure 1: (**a**) A traditional neural network pipeline which is enhanced with external knowledge to handle GLUE tasks such as entailment, similarity, and other types of natural language inference tasks. Initially, the tokenized text undergoes encoding by a language model, which outputs an embedding. Following that, a method based on graph embedding is employed to extract and embed a subgraph that is relevant to the input text. This involves extracting entities within a certain distance threshold from the entities present in the text. Subsequently, the two embeddings - the language model embedding and the graph embedding, are merged and passed through a classification head to obtain the predicted logits. To train this model, the cross-entropy loss between the predicted logits and the actual output is minimized. (**b**) The RDR paradigm. The tokenized input goes through a paraphrasing model, and a paraphrasing loss is calculated. Additionally, the graph-embedding-based subgraph extraction method is compared against a ground truth subgraph, then a graph embedding loss is computed. The total loss is the sum of the losses from the paraphrasing loss, graph embedding loss, and classification head loss.

to the predefined train-validation split established by the GLUE benchmark. The reported results (Table 1) represent the average of two independent runs. The knowledge graphs utilized in this study include DBPedia, ConceptNet, Wiktionary, WordNet, and the OpenCyc Ontology. These knowledge graphs consist of interconnected objects and their relationships, forming semantic associations (Auer et al. 2007; Speer, Chin, and Havasi 2017; Matuszek et al. 2006). Figure 2 illustrates the process of extracting subgraphs from the input text.

Approximately 300K subgraphs are obtained from the knowledge graphs across all inputs. The relationships include Antonym, DistinctFrom, EtymologicallyRelatedTo, LocatedNear, RelatedTo, SimilarTo, Synonym, AtLocation, CapableOf, Causes, CausesDesire, CreatedBy, DefinedAs, DerivedFrom, Desires, Entails, ExternalURL, FormOf, HasA, HasContext, HasFirstSubevent, HasLastSubevent, HasPrerequisite, HasProperty, InstanceOf, IsA, MadeOf,

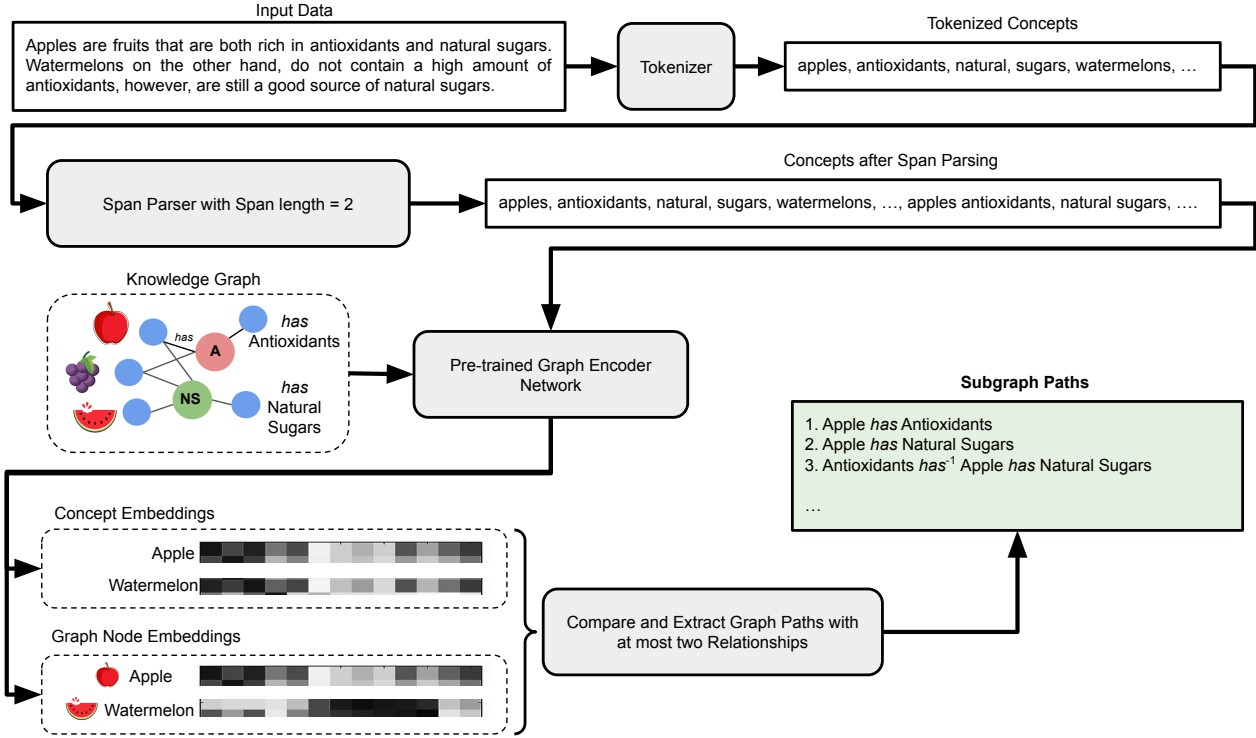

Figure 2: Illustration of the process of extracting subgraphs from the knowledge graph given an input instance. For the pre-trained Graph Encoder Network, we use ConceptNet's Numberbatch embeddings and use a span length of three in our experiments.

MannerOf, MotivatedByGoal, ObstructedBy, PartOf, ReceivesAction, SenseOf, SymbolOf, and UsedFor.

We experiment with the best-performing models from huggingface with $\leq$ 500 million parameters, i.e., BERT, RoBERTa, and ALBERT. We note that RoBERTa is especially suited for entailment tasks (Devlin et al. 2018; Liu et al. 2019; Lan et al. 2019). Table 1 shows the accuracies for all the models with and without the RDR training method. For the graph embedding model in the RDR model, we use the TransE algorithm (Bordes et al. 2013). The RDR method shows significant improvements over the baselines, with just 10% of the knowledge graph. We choose a random 10% of the knowledge graph triples for each training run to avoid the observed phenomenon of gaming of evaluation benchmarks by language understanding models, i.e., to avoid the model fitting spurious patterns in the additional knowledge as a means to achieve high accuracy scores.

## Conclusion and Future Work

This paper presents the formalization and initial experimental outcomes of a new training approach known as "Recap, Deliberate, and Respond" (RDR). We demonstrate that RDR achieves superior performance compared with baseline methods. RDR also shows resilience against manipulation of benchmarks (as evidenced by observing performance improvements when using only a random 10% of the available knowledge during each training iteration). Subsequent

| M | MNLI-M | MNLI-MM | QNLI | QQP | WNLI | MRPC | RTE |
|---|---|---|---|---|---|---|---|
| B | 82.44 | 83.52 | 90.49 | 90.1 | 54.92 | 82.11 | 66.06 |
| $R_B$ | 83.31 | 84.47 | 91.25 | 90.78 | 56.33 | 82.86 | 67.87 |
| R | 85.07 | 85.19 | 91.06 | 90.17 | 56.33 | 86.03 | 62.81 |
| $R_R$ | 85.78 | 85.95 | 91.85 | 90.96 | 57.75 | 86.76 | 63.53 |
| A | 84.34 | 85.32 | 90.6 | 90.25 | 57.75 | 86.27 | 66.06 |
| $R_A$ | 85.03 | 85.82 | 91.18 | 90.74 | 59.15 | 87 | 66.43 |

Table 1: **$R_B$**: RDR$_B$, **$R_A$**: RDR$_A$, **$R_R$**: RDR$_R$, **M**: MODEL, **MNLI-M**: MNLI-MATCHED, **MNLI-MM**: MNLI-MISMATCHED, B: BERT-BASE, R: ROBERTA-BASE, A: ALBERT-BASE-V2. Results for RDR method compared to models that do not use the RDR method. We see improvements of up to 2%, on average 1% using only 10% of the knowledge graphs triples showing promise of the RDR methodology for improved language understanding.

research will explore the utilization of diverse knowledge sources, including domain-specific knowledge, broader general knowledge (e.g., from Wiki, Unified Medical Languaging System), and others. We will also experiment using large SOTA models (e.g., LLMs such as mistral, llama, falcon, and ChatGPT) and diverse geometrical embeddings (e.g., CompIEx, HolE, DistMult) within the RDR framework.

## Acknowledgements

This research is built upon prior work (Zi et al. 2023; Rawte et al. 2022; Roy et al. 2023c; Venkataramanan et al. 2023; Roy et al. 2023a, 2022, 2023a), and supported in part by NSF Award 2335967 "EAGER: Knowledge-guided neurosymbolic AI" with guardrails for safe virtual health assistants". Opinions are those of the authors and do not reflect the opinions of the sponsor (Roy et al. 2021, 2023b; Sheth et al. 2021, 2022; Sheth, Roy, and Gaur 2023).

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
