# OpenReview forum: "RDR: The Recap, Deliberate, and Respond Method for Enhanced Language Understanding"
_AAAI.org/2024/Workshop/NuCLeaR — NuCLeaR 2024_

### Official Review · Reviewer_RYyf · 2023-12-07
**The paper introduces a novel Neuro-symbolic training approach to enhance Natural Language Understanding (NLU) models. It claims to improve performance without manipulating benchmarks. The review suggests a clearer alignment between research question and methodology, a more detailed literature review, and further evidence for claims. The writing is praised for clarity, and the proposed approach is really promising, but it needs more work.**

**Rating:** 6
**Confidence:** 4

**Review:**

## 1. ****General Description of the paper****

This paper falls in the category of the first day of the workshop : Neuro-symbolic methods. It introduces a novel approach to train and enhance the Natural Language Understanding (NLU) models that are known to use knowledge graphs during training. The proposed method claims to increase performance without manipulating NLU benchmarks.

## 2. **************************************************Evaluation of the quality**************************************************

For the sake of this review, these indicators of research quality were inspected :

- **Do hypotheses follow logically from previous work?**

    A clearer alignment between the research question and the methodology is needed. The authors address the vulnerability of NLU benchmarks to manipulation,  but evaluate their proposed approach on the same benchmarks they criticize.

- **Are the background literature and study rationale clearly articulated?**

    While the authors mentioned some related literature, they haven’t included a well-detailed section for related or previous work, which is needed to clarify the differentiation of the proposed approach from previous methods in terms of combining input data with external knowledge, and how their method doesn’t manipulate bechmarks to inflate performance.


- **Could this methodology have answered the addressed issue/question?**

    The problematic being addressed is the vulnerability of NLU benchmarks to manipulation and the difficulty of evaluating NLU models, so the reader might conclude that the paper is going to propose new ways of fairly evaluating NLU models, and not propose a whole new NLU pipeline. The authors are highly suggested to find alternative problems to address that could emphasize their approach's ability to avoid benchmark gaming.


**Recommendation:** A more thorough literature review is encouraged, as it can significantly emphasize the work of the authors and highlight the potential of their proposed method.

## 3. ************************************************Evaluation of clarity :************************************************

- The style of writing as well as the language being used by the authors is clear and straight-forward.
- The authors guide readers throughout the whole article and explain every figure.
- The paper is formatted according to AAAI conference guidelines, it respects all the formatting instructions.
- The figures are really well presented and commented.

## 4. **Evaluation of originality:**

- The idea of incorporating additional objectives in the training process and minimizing a combined loss to enhance the training has been used in many domains (in Physics Informed Neural Networks for example), but I am not aware whether this idea as been applied before in the context o NLU models.
- The incorporation of the new losses to initial cross entropy sounds promosing in terms of helping the model learn to authentically understand natural language and perform better on NLU tasks, but this has to be further evaluated and confirmed.

## 5. **********Evaluation of the significance of the work**********

- According to their result section, they got an improvement of up to 2% and average of 1% compared to baselines : BERT, RoBERTa and ALBERT on GLUE benchmarks (actually the maximum is 1.81% and average is approximately 0.67%, after redoing the calculation from Table 1).
- The proposed method of training is promising, but lacks more work.

## 6. ****Critical review of each section****

Instead of a general pros and cons list, here are bullet points with critical reviews of each section individually :

**Abstract :**

- The discussed issue is the vulnerability of NLU benchmarks to manipulation by neural models, so proposing a new method and testing it on these same benchmarks raise skepticism about its effectiveness.
- The claim of avoiding benchmark gaming needs more substantial evidence.

**Introduction**

- There is some redundancy in the first two sentences, they said the same thing twice by reformulating it differently.
- When mentioning previous works, they mentioned a very recent approach  (Zhu et al. 2023) with under 20 citations and said it’s a well established approach, and based their addressed issue on it.
- They mentioned that the approach they used for training prevents models from exploiting statistical artifacts to artificially inflate performance metrics, but they didn’t argument.
- The promised examples illustrating the capacity of models trained using the RDR methodology are missing in the results section.



**RDR methodology**

- The traditional method that they described lacks references or examples.
- They haven’t elaborated on the difference between the language model in the traditional method and the paraphrasing model in their proposed method.

**Figure 1 :**

- The process of extracting the ground truth subgraph from the input and external graph, in order to compute the graph embedding loss, is not clearly explained.
- No details were provided on its inner workings of the paraphrasing model : it outputs recapped input and embedded input at the same time.

**Experiments and Results :**

- Mention of five knowledge graphs with citation for only three.
- Comparison was done with baseline models that do not follow the traditional method described in the methodology section, so the methods are totally different. For example : BERT doesn’t use a knowledge graph in training.
- Evaluation on benchmarks criticized for vulnerability and manipulation without substantial arguments.
- The claim of non-manipulation based on randomly choosing 10% of the knowledge graph lacks supporting details.

---

### Official Review · Reviewer_SAf5 · 2023-12-08
**Review of RDR**

**Rating:** 5
**Confidence:** 3

**Review:**

**Summary**

This paper proposes a modified approach for NLU tasks called Recap Deliberate and Respond (RDR). Authors consider a traditional neural network approach that just combines embedding of the natural language input with an emdedding of relevant external subgraph and train a classifier model on top of this with cross entropy losses. The proposed RDR approach modifies this by adding two additional losses to the final cross entropy loss - a) paraphrase loss on text embedding b) graph embedding loss on graph embedding.

Experimental results on GLUE benchmark shows the proposed method slightly outperforms traditional approach on BERT, ROBERTA and ALBERT models

**Strengths**

1. Using paraphrasing loss and graph embedding loss for NLU is interesting
2. This paper is well organised and reasonably clear

**Weaknesses**

1. Limited experimental results and the improvement is less than 1%. Comparisons with other SOTA methods for GLUE will be helpful
2. Its not very clear how RDR addresses the limitations of existing works discussed in Introduction

**Questions**

1. How does the Language Model in traditional method differs from the Paraphrasing Model in RDR?

---

### Decision · Program_Chairs · 2023-12-11

Accept